nanotechnology

AgNPs, polyol method, sinter, microwave, conductivity

**Author for correspondence:**
Xingzhong Guo
e-mail: msewj01@zju.edu.cn

# Facile preparation of high conductive silver electrodes by dip-coating followed by quick sintering

Tianrui Chen[1], Hui Yang[1,2], Shengchi Bai[1], Yan Zhang[1] and Xingzhong Guo[1]

[1]State Key Laboratory of Silicon Materials, School of Materials Science and Engineering of Zhejiang University, Hangzhou, 310027, People's Republic of China
[2]Zhejiang-California International Nanosystems Institute, Zhejiang University, Hangzhou 310058, People's Republic of China

XG, 0000-0003-2901-971X

With polyol-synthesized silver nanoparticles (AgNPs) as raw materials, the silver electrodes with high conductivity were fabricated via a dip-coating method followed by sintering process, and the effects of the dip-coating and sintering process on the conductivity and surface roughness of silver electrodes were investigated in detail. The silver film with a thickness of 1.97 µm and a roughness of about 2 nm can be prepared after dip-coating at a pulling rate of 500 µm s$^{-1}$ for 40 coating times. The non-conductive dip-coated silver films are transformed into conductive silver electrodes after conventional sintering in a muffle oven, infrared sintering and microwave sintering, respectively. Compared with high sintering temperature and long sintering time of conventional sintering and infrared sintering, microwave sintering can achieve quick sintering of silver films to fabricate high conductive silver electrodes. The silver electrodes with a sheet resistance of 0.75 Ω sq$^{-1}$ and a surface roughness of less than 1 nm can be obtained after microwave sintering at 500 W for 50 s. The adjustable dip-coating method followed by quick microware sintering is an appropriate approach to prepare high conductive AgNPs-based electrodes for organic light-emitting diodes or other devices.

## 1. Introduction

In recent years, organic light-emitting diodes (OLED) have attracted considerable attention owing to their advantages such as low power consumption, fast response times, good colour gamut, eco-friendly nature, self-illumination and so on [1–5]. For OLED, a lot of research has focused on its transparent and

This article has been edited by the Royal Society of Chemistry, including the commissioning, peer review process and editorial aspects up to the point of acceptance.

conductive anodes [6–10], while little attention is paid to the cathode. At present, vacuum evaporation is the most common method for the preparation of silver or aluminium thin-film electrodes or other films [11–13]. As is well known, the thin-film electrodes prepared by vacuum evaporation possess high conductivity, good smoothness and extremely thin thickness; however, vacuum evaporation is also limited owing to insurmountable shortcomings such as expensive equipment, complicated vacuuming processes and low economic efficiency. Therefore, it is very urgent to find an all-solution method to construct silver electrodes for cathodes of OLED or other electronic devices [14–17].

There are many methods used to prepare silver electrodes, such as solution self-assembly and ink-jet-printing, which are suitable for replacing vacuuming evaporation plating. Ishwor Khatri *et al.* [18] prepared silver electrodes by self-assembled silver nanowires (AgNWs) to construct organic–inorganic hybrid solar cells. The self-assembled AgNWs mesh electrodes show low sheet resistance ($8 \, \Omega \, sq^{-1}$) with enhanced transparency in the ultraviolet and infrared regions. Yu Chen-Chiang *et al.* [19] introduced a porous silver thin-film cathode fabricated by a simple ink-jet-printing process for low-temperature solid oxide fuel cell applications. However, the products obtained from these studies still have defects such as high sheet resistances and strict requirements to the precursors.

In this work, we introduce a facile preparation approach of the silver electrodes, including using synthesized silver nanoparticles (AgNPs) by a polyol method as raw materials, coating silver film by a dip-coating method and sintering the silver film by quip sintering. The dip-coating, sintering processes and sintering mechanism of AgNPs for constructing silver electrodes are discussed. The resultant AgNPs-based electrodes showed good smoothness and excellent conductivity (less than $1 \, \Omega \, sq^{-1}$), which could be applied as the cathode to construct OLED, solar cells or other electronic devices.

# 2. Experimental procedure

## 2.1. Chemicals and materials

Silver nitrate ($AgNO_3$; Aladdin Ind. Co., China, 99%) was used as the silver source. Ethylene glycol (EG; Aldrich, 99.8%) was the reducing agent as well as the solvent. Polyvinylpyrrolidone (PVP; $M_w = 58000$, Aladdin Ind. Co., China) was used as the stabilizer as well as the anisotropic agent. Anhydrous copper chloride ($CuCl_2$; Aladdin Ind. Co., China, 98%) was used as the ion additive. Acetone (PA; Aldrich, 99.5%) was used as the washing solvent. All chemicals were used without further purification.

## 2.2. Preparation of silver nanoparticles

In the synthesis process, 50 ml EG was preheated in an oil bath at 120°C for 1 h to remove the water, and 5.0 g PVP was added into the solvent and dissolved to form a polyol solution. Twenty-five millilitres $FeCl_3$ EG solution (600 µM) was added into the polyol solution, and subsequently, 25 ml $AgNO_3$ EG solution (600 mM) was added into the mixed solution dropwise within 3 min. Vigorous stirring was maintained throughout the entire process. After 30 min, the one-pot reaction was quenched by cooling. The cooled solution was centrifugal at 5000 r.p.m. with PA for at least 30 min until the suspension sank to the bottom and the up-solution was clear. Then the precipitate was centrifuged for three times at 3000 r.p.m. for 10 min each time and washed with ethanol to remove excess solvent, PVP and other impurities in the supernatant.

## 2.3. Fabrication of silver electrodes

Silver electrode films were prepared by dip-coating on $2 \times 2$ cm glass slides with a dip coater (SYDC-100, Shanghai San Yan Technology Corp., Ltd). These substrates were thoroughly cleaned with detergent and washed with deionized water, sonicated in ethanol for 5 min and dried in an oven. Then the substrates were clamped to the fixture above the ethanol solution of AgNPs. After setting the cycles and the rates of pull-ups and other parameters, the dip coater was used to prepare silver film.

Three sintering processes, that is, conventional sintering, infrared sintering and microwave sintering, were used to achieve the sintering of the dip-coated silver films after dip-coating. In the conventional sintering, the silver films were sintered in a muffle furnace at different temperatures (50–400°C). In the infrared sintering, the silver films were placed under an infrared baking lamp and baked at different powers (100–250 W) for different sintering times (30–60 min). In the microwave sintering, the silver

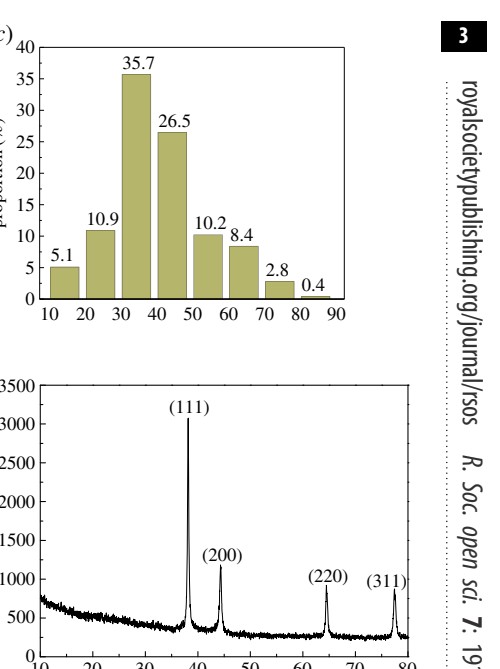

**Figure 1.** TEM of AgNPs magnified (*a*) 20 000×; (*b*) 50 000×; (*c*) size statistics of AgNPs; (*d*) EDS diagram and element composition of AgNPs; (*e*) XRD pattern of AgNPs indicating the fcc structure of silver.

films were placed in a microwave processing instrument at different microwave powers (250–1000 W) for different sintering times (20–50 s). All three sintering processes are in air atmosphere.

## 2.4. Characterization

Morphologies of the AgNPs and films were characterized by using a scanning electron microscope (SEM; SU8010, Hitachi Ltd, Japan) and a transmission electron microscope (TEM; JEM-1230, JEOL Ltd, Japan). Powder X-ray diffractions (XRD) of the AgNPs were performed with an Empyrean 200895 diffractometer (XRD-6000, Shimadzu, Japan) using CuK ($\lambda$ = 0.154 nm) radiation as an incident beam. The chemical composition of the AgNPs was obtained using energy-dispersive X-ray spectroscopy (EDS). The sheet resistance of the silver films was measured using a 4-point probe sheet resistance meter (RTS-9, 4 PROBES TECH). The thickness and roughness of the silver films were carried out on a step profiler (DEKTAK-XT, Bruker).

# 3. Result and discussion

## 3.1. Morphology and microstructure of silver nanoparticles

In this work, AgNPs were synthesized by a common polyol process [20–27], and the morphology and particle size of as-prepared AgNPs were investigated and are shown in figure 1*a*–*c*. The diameters of 1000 Ag particles were measured and counted by Nano Measurer software. It is demonstrated that the synthesized AgNPs have a small particle size and narrow particle size distribution, the average size of AgNPs is about 43 nm, and the particle size of about 62.2% is distributed between 32 and 54 nm. The composition of AgNPs was characterized by EDS and XRD, as shown in figure 1*d*,*e*. It can be seen that the content of silver reaches 83.77%, the contents of C and N elements are 13.69% and 2.55%, respectively, indicating the existence of PVP in the samples. AgNPs coated with PVP can be uniformly dispersed in ethanol owing to the unit group (N=C–O) of PVP. The N=C–O group has a pair of lone pair electrons, and the electrons will combine with the silver ions, which promotes the combination of PVP and silver atoms to form the complex and also improves the dispersion of AgNPs [26]. It is observed from the XRD pattern of the AgNPs that all the reflection peaks can be indexed to silver, which confirms that the as-prepared particles are pure AgNPs. The lattice constant calculated

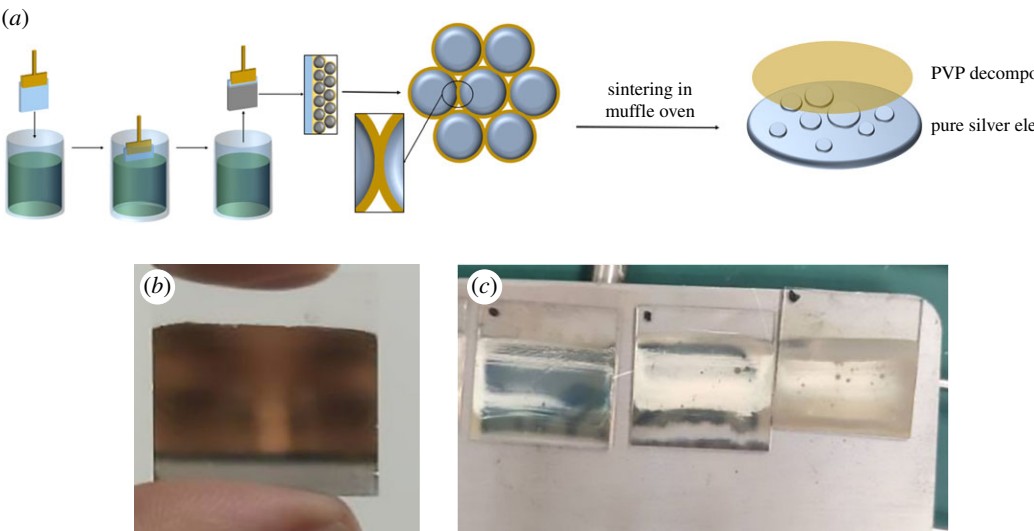

**Figure 2.** (a) Process of dip-coating and conventional sintering of silver film; (b) photographs of silver film without conventional sintering; (c) photographs of silver films after conventional sintering in a muffle oven.

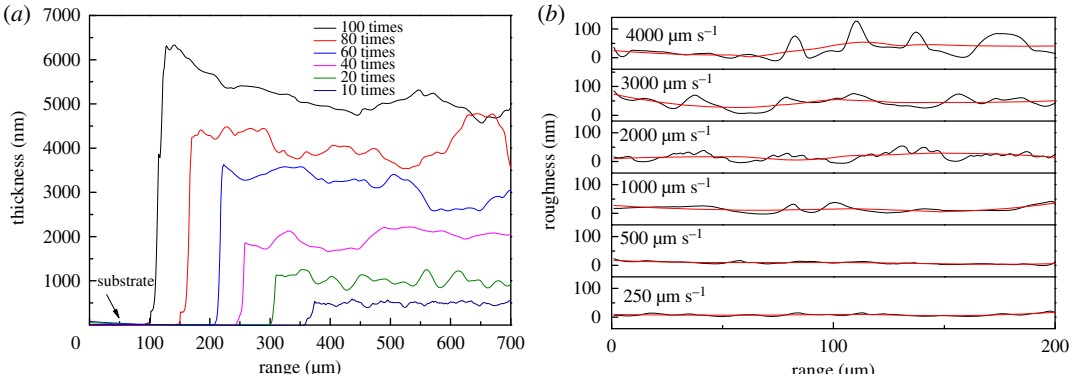

**Figure 3.** (a) Thickness of silver films coated at different coating times; (b) roughness of silver films coated at different pulling rate.

from the pattern is 4.087 Å, which is consistent with the standard value of 4.086 Å (JCPDS Card File No. 4-783) [28]. It indicates that there are no impurities detected from the AgNPs.

## 3.2. Dip-coating and conventional sintering preparation of silver electrodes

There are many ways to prepare thin films, such as spinning, scraping, dip-coating and so on [29–34]. Among these methods, the dip-coating method is a simple way to fabricate smooth thin film by repeated coating many times [35–37]. In this work, dip-coating is adopted to prepare AgNP-based thin film, and the process of dip-coating followed by conventional sintering is shown in figure 2a. After dip-coating, the dip-coated silver electrodes without any post-treatment are non-conductive, which indicates that the silver particles are coated by PVP. As shown in figure 2b, the dip-coated thin film becomes very bright and smooth and looks like a mirror. After conventional sintering in muffle furnace, the PVP is decomposed, and silver particles melt and coagulate together to form conductive electrodes of pure silver. The as-prepared silver electrodes are rough and become a little white.

The thickness and roughness of dip-coated silver films are measured by a step profiler, as shown in figure 3. The calculation of roughness was carried out in the supporting software of the step profiler. It can be seen that the thickness of silver films increases with the increase of coating times. When the number of coating times increase from 10 to 100 times, the thickness of silver films increases from 498 nm to 5.19 µm. The roughness of silver films becomes larger with the increase of pulling rate. The roughness of silver films is 2.25 nm at 250 µm s⁻¹ of pulling rate, and becomes 1.98 nm when the pulling rate increases to 500 µm s⁻¹. With a further increase of pulling rate, the roughness rapidly

**Table 1.** Conductivity of silver electrodes at different coating times and heating temperatures.

| sintering temperature (°C) | no. of coating times | | | | | |
|---|---|---|---|---|---|---|
| | 10 | 20 | 40 | 60 | 80 | 100 |
| without sintering | — | — | — | — | — | — |
| 50 | — | — | — | — | — | — |
| 100 | — | — | 4 | 2.4 | 1.9 | 0.7 |
| 150 | — | 10 | 2.1 | 1.6 | 1.2 | 0.5 |
| 200 | — | 4 | 1.4 | 0.95 | 0.86 | 0.44 |
| 250 | 49.6 | 1.9 | 0.94 | 0.71 | 0.57 | 0.23 |
| 300 | — | — | 3.9 | 0.5 | 0.45 | 0.16 |
| 400 | — | — | — | — | — | — |

increases, and reaches 18.28 nm at 4000 µm s$^{-1}$. Considering the structure of OLED or other devices, the suitable silver film with a thickness of 1.97 µm and a roughness of about 2 nm can be prepared after dip-coating at 500 µm s$^{-1}$ of pulling rate for 40 coating times.

The conventional sintering of dip-coated silver films was carried out in a muffle furnace to prepare conductive silver electrodes. Table 1 shows the conductivity of silver films with different numbers of coating times and after being conventionally sintered at different sintering temperatures for 30 min. It is noted that the dip-coating times and conventional sintering have an important effect on the conductivity of silver films. The silver films without sintering and sintered at 50°C are non-conductive, although they are dip-coated many times. When the sintering temperature is 100°C, the silver films dip-coated 10 or 20 times are still non-conductive, while the silver films become good conductors, after being dip-coated more than 40 times. When the sintering temperature increases from 100°C to 350°C, most of silver films with different coating times are conductive, while the silver films become non-conductive again, when the sintering temperature rises to 400°C. It can be seen from the conductive silver films at a constant sintering temperature that the conductivity of silver films increases with an increase in number of dip-coating times, and at constant number of dip-coating times, the conductivity of silver films increases with the rising of sintering temperature. After being dip-coated for 40 times and conventionally sintered at 250°C for 30 min, a silver film with a low sheet resistance (less than 1 Ω sq$^{-1}$) can be obtained.

The surface morphology of the silver electrodes after conventional sintering was characterized by SEM, shown in figure 4. It can be noted that in the silver film without sintering, the nanoparticles are piled up on the substrate, there are many pores between the AgNPs. When the sintering temperature reaches 100°C or 200°C, AgNPs are still piled up, with a few melted nanoparticles. When the sintering temperature rises to 300°C, all the nanoparticles melt together to form an individual layer, while the graininess can be still observed. When the temperature further increases to 400°C, the nanoparticles are completely deformed and wrecked, and there exist some isolated large silver particles or little silver bulk. TG and DTA tests are carried out on the AgNP powder at a heating rate of 10°C min$^{-1}$ in the atmosphere. TG records the weight change of the sample, while DTA records the energy change of the sample. It is seen from the TG and DTA of AgNPs (figure 4$f$) that there exists a small exothermic peak at around 200°C, which indicates that the nanoparticles begin sintering at this temperature. AgNPs containing high energy atoms are highly unstable, and will diffuse and cause surface sintering at high temperature. A sharp exothermic peak near 300°C is apparently attributed to the decomposition of PVP, which suggests that the temperature of conventional sintering should be designed to more than 300°C to remove the PVP. It can be seen from the TG diagram that there is a weight increase before 200°C, which may be caused by oxidation or instrument error. An obvious weight loss between 200°C and 400°C is owing to the decomposition of PVP.

## 3.3. Infrared sintering and conductivity of silver electrodes

Infrared sintering technology is a photonic sintering approach for metal nanoparticles [38–40]. In this work, the infrared sintering process is used to promote the deformation of PVP on the surface of AgNPs and improve the conductivity of the silver film. The schematic and physical drawings are shown in figure 5$a$. The silver electrode is directly irradiated and sintered by infrared light from the

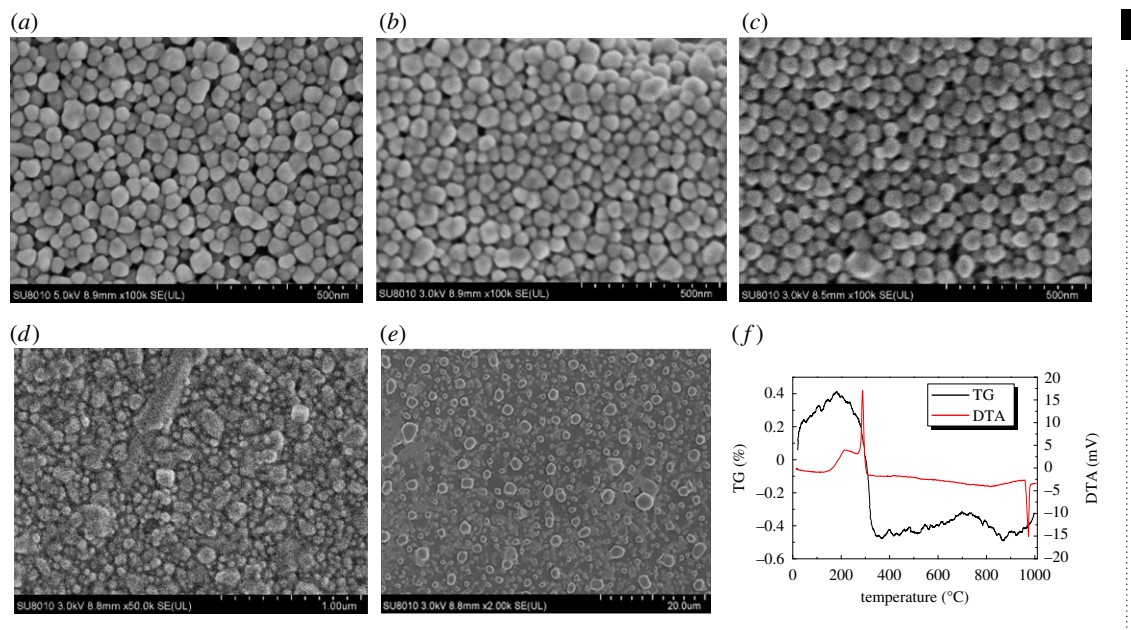

**Figure 4.** SEM image of silver films without sintering (*a*) and sintering at 100℃ (*b*), 200℃ (*c*), 300℃ (*d*) and 400℃ (*e*); (*f*) DTA and TG curve of AgNPs.

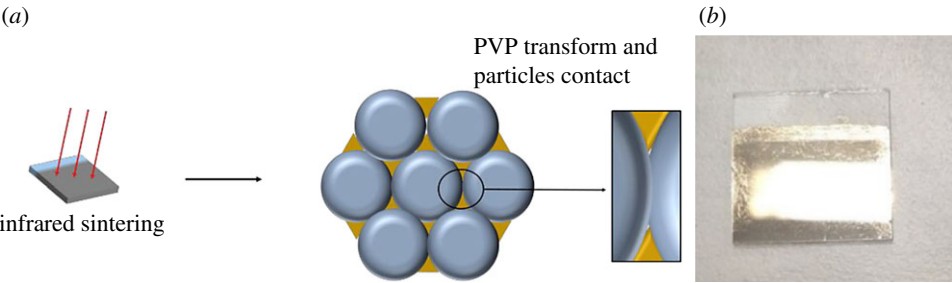

**Figure 5.** (*a*) Process schematic diagram and (*b*) photograph of silver film after infrared sintering.

top side. Infrared light does not bring enough energy to melt the AgNPs and decompose the PVP. Therefore, silver particles can be in close contact, so silver electrodes have better conductivity and flatness. It can be seen from figure 5*b* that the silver film still reflects as smoothly as a mirror.

Figure 6 is the sheet resistance of the silver film obtained with different baking times and power. Under the same conditions of baking time of 60 min, the sheet resistance of the silver film decreases with the increase of power. The silver film after being baked at 100 W is relatively high, at 27.6 $\Omega$ sq$^{-1}$. When the power increases to 150 W, the sheet resistance decreases to 8.65 $\Omega$ sq$^{-1}$, which is still a high value. With the increase of power to 200 W, the sheet resistance continues to decrease to 1.86 $\Omega$ sq$^{-1}$, a lower value. Further increasing power to 250 W, the sheet resistance becomes very low, at 0.89 $\Omega$ sq$^{-1}$. Under the same condition of power of 200 W, the sheet resistance of the silver film decreases with the increase of baking time. When the baking time is 30 min, the sheet resistance is 12.31 $\Omega$ sq$^{-1}$, and decreases to 2.09 $\Omega$ sq$^{-1}$ when the baking time increases to 40 min. With a further increase in baking time to 50 min, the sheet resistance continuously decreases to 1.7 $\Omega$ sq$^{-1}$, and slightly increases to 1.86 $\Omega$ sq$^{-1}$ at the baking time of 60 min. With the increase of irradiating power and time of the infrared sintering, the conductivity of silver film increases and the sheet resistance declined, this results from the deformation effect of PVP.

Figure 7 shows SEM images of silver electrodes after being baked at different baking powers for 30 min and baked at 200 W for various baking times, respectively. It can be seen that the baking power and baking time do not affect the surface morphology of the silver films, and the nanoparticles do not obviously melt.

Figure 8 shows the thickness and roughness of infrared sintered silver electrodes measured by a step profiler. The thicknesses of silver films after infrared sintering at 50 W for 30 min, 200 W for 30 W, and

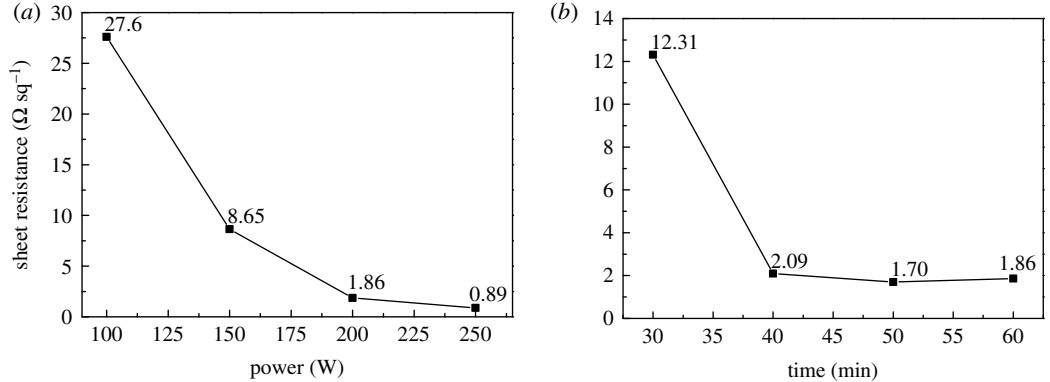

**Figure 6.** Sheet resistance of silver electrodes at different baking powers (*a*) and baking times (*b*) by infrared sintering.

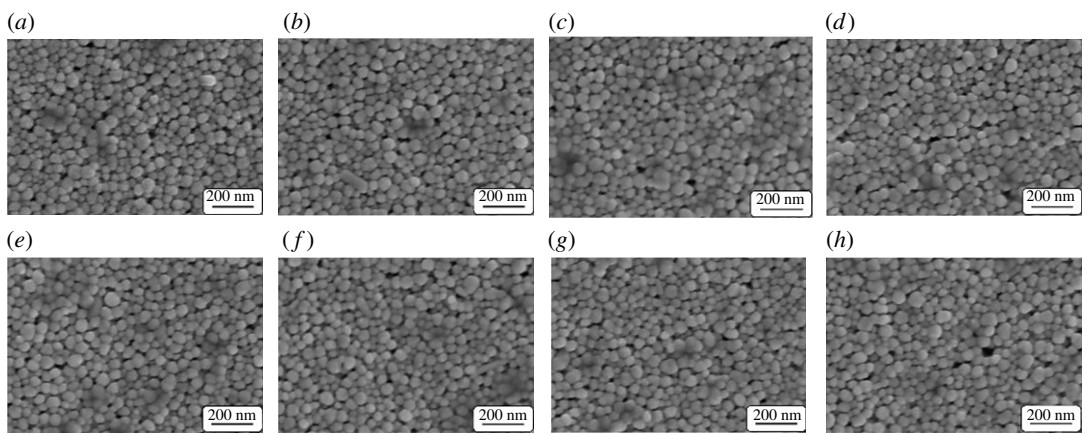

**Figure 7.** SEM images of silver electrodes after being baked at (*a*) 100 W, (*b*) 150 W, (*c*) 200 W and (*d*) 250 W power for 30 min, respectively, and baked at 200 W for (*e*) 30 min, (*f*) 40 min, (*g*) 50 min and (*h*) 60 min, respectively.

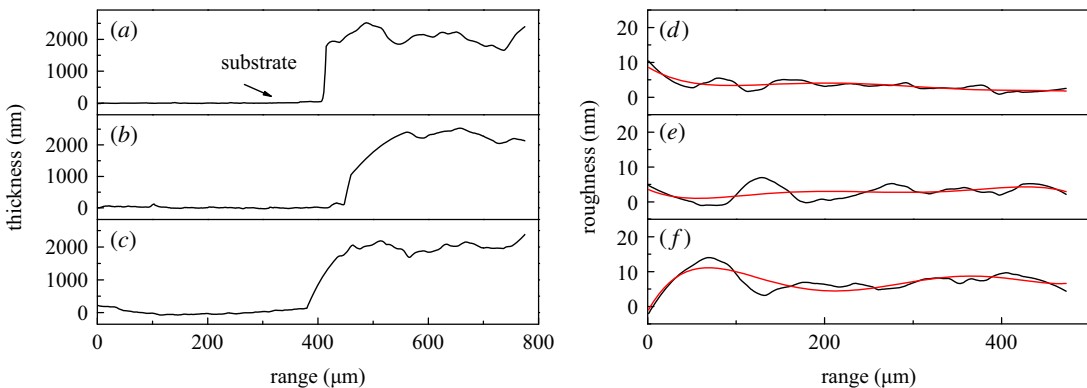

**Figure 8.** Thickness and roughness of silver electrodes baking at (*a,d*) 50 W for 30 min; (*b,e*) 200 W for 30 min; (*c,f*) 200 W for 60 min.

200 W for 60 min are 1.94, 2.16 and 2.07 µm, and the roughness of the three samples is 1.3, 1.39 and 0.68 nm, respectively, indicating no significant change after infrared sintering. It can also be considered that infrared sintering has no obvious effect on the surface morphology of the silver film, while promoting the deformation of PVP to increase the conductivity of silver film.

## 3.4. Microwave sintering and conductivity of silver electrodes

Microwave sintering is widely used for the densification of ceramic materials and in synthetic chemistry because of its advantages such as uniform, fast and volumetric heating [41,42]. In this work, the microwave sintering process is also used to promote the deformation of PVP on the surface of AgNPs

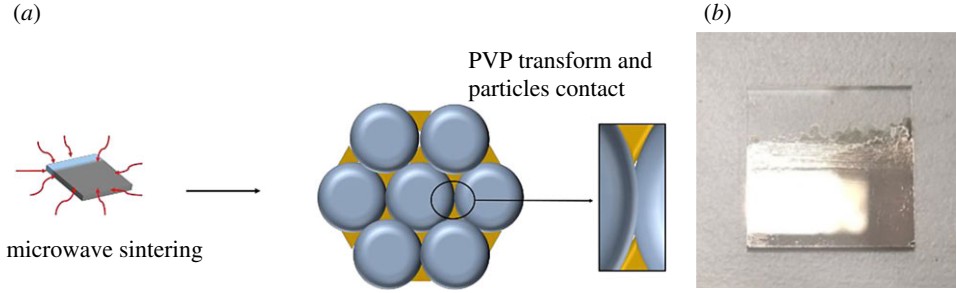

**Figure 9.** (a) Process schematic diagram and (b) photograph of the silver film after microwave sintering.

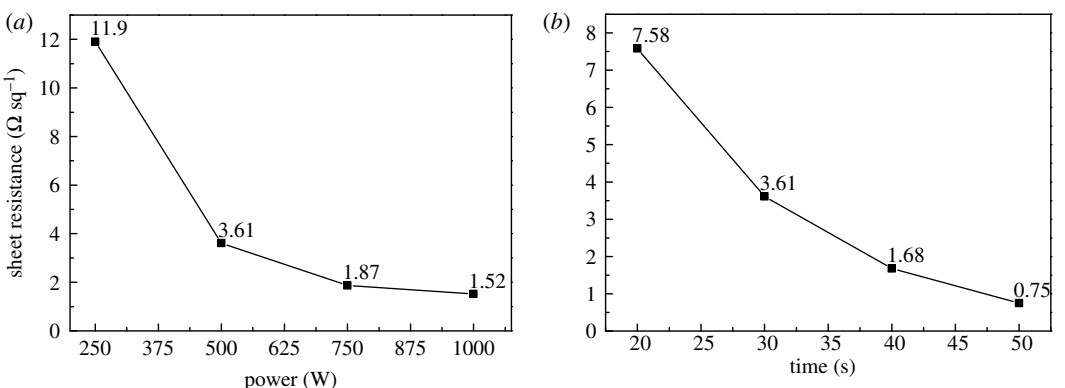

**Figure 10.** Sheet resistance of silver electrodes at different baking powers (a) and baking times (b) by microwave sintering.

and improve the conductivity of the silver film. Its schematic process diagram and physical photo diagram are shown in figure 9. The silver electrode is irradiated and sintered by microwaves in all directions, and all the nanoparticles on the surface and inside of the silver electrode will be heated. The metals can reflect the microwaves, and the PVP in silver film can be heated strongly. During microwave sintering, AgNPs do not melt under low power microwave, and the PVP is deformed by microwave energy to make silver particles contact with each other. Sintering time is very short because of the high penetration and high energy of microwaves, and the reflection of AgNPs. The characteristics of microwave sintering is uniformity, sufficiency and rapidity.

Figure 10 shows the sheet resistance of silver electrodes obtained under different microwave sintering times and powers. The sheet resistance of silver electrodes decreases with the increase of microwave power at a constant sintering time, and decreases with increasing sintering time at constant microwave power. At 500 W for 50 s, the sheet resistance of silver electrodes is the lowest, at 0.75 $\Omega$ sq$^{-1}$. The microwave sintering is similar to the infrared sintering, but the temperature field is more uniform because of heating from all directions. With the increase of the power and time of microwave sintering, the conductivity of silver electrodes increases and the sheet resistance declines.

Figure 11 shows the SEM images of silver electrodes after microwave sintering at 250, 500, 750 and 1000 W for 30 s and microwave sintering at 500 W for 20, 30, 40 and 50 s, respectively. It can be also seen that the microwave sintering power and time have no obvious effect on the surface morphology of the silver electrodes.

Figure 12 shows the thickness and roughness of microwave sintered silver electrodes measured by a step profiler. The thicknesses of silver electrodes after microwave sintering at 250, 1000 and 500 W for 50 s are 2.18, 1.81 and 2.54 µm respectively, and the roughness are 1.16, 0.91 and 0.90 nm respectively. It confirms that microwave sintering has no obvious effect on the surface morphology of the silver electrodes.

The sheet resistances of silver electrodes after suitable conventional sintering, infrared sintering and microwave sintering are compared and shown in figure 13a. It can be seen that the sheet resistance of silver electrodes after microwave sintering is the lowest among the three sintering methods, at 0.75 $\Omega$ sq$^{-1}$. A silver electrode obtained by microwave sintering not only has good conductivity, but also has very short sintering time (less than 1 min). Figure 13b shows a verification circuit consisting of a power supply, a switch and a small light bulb. The lighting of the bulb proves that the microwave sintered silver electrode has good conductivity.

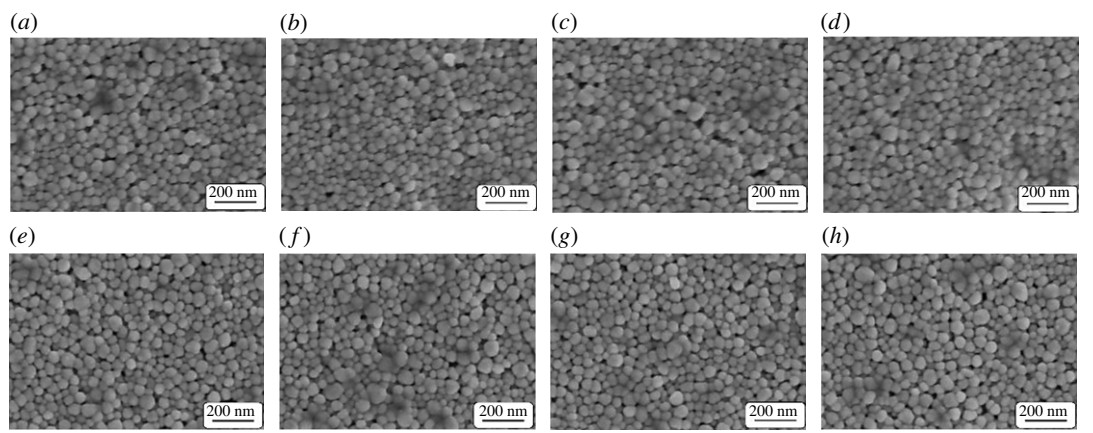

**Figure 11.** SEM images of silver electrodes after microwave sintering at (*a*) 250 W, (*b*) 500 W, (*c*) 750 W and (*d*) 1000 W for 30 s, respectively, and microwave sintering at 500 W for (*e*) 20 s, (*f*) 30 s, (*g*) 40 s and (*h*) 50 s, respectively.

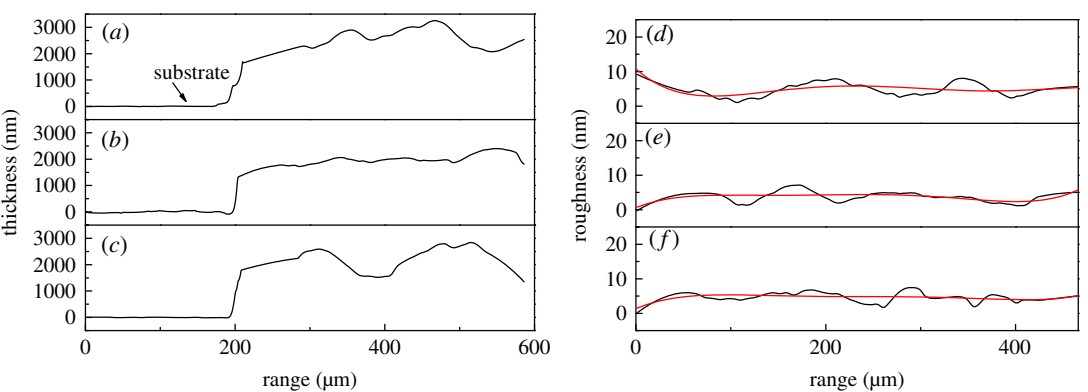

**Figure 12.** Thickness and roughness of silver electrodes after microwave sintering at (*a,d*) 250 W for 30 s; (*b,e*) 1000 W for 30 s; and (*c,f*) 500 W for 50 s.

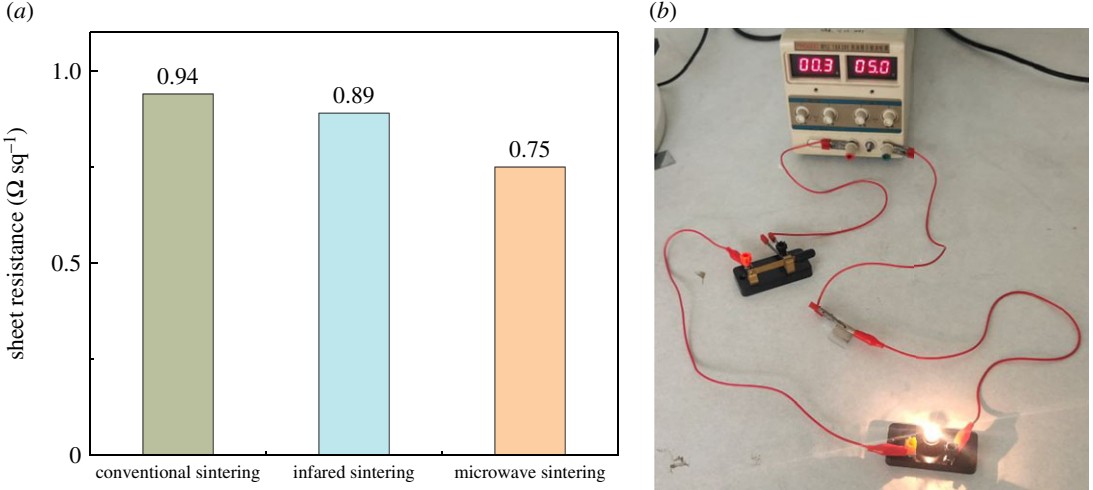

**Figure 13.** (*a*) Conductivity comparison among silver electrodes by conventional sintering at 250°C for 30 min, infrared sintering at 250 W for 30 min, and microwave sintering at 500 W for 50 s; (*b*) photograph of verification circuit to prove the conductivity of microwave sintered silver electrodes.

## 4. Conclusion

In summary, the silver electrodes were facilely prepared by using synthesized AgNPs by a polyol method as raw materials, coating silver film by the dip-coating method and sintering the silver film by a sintering

process. The surface morphology and conductivity of dip-coated silver film and silver electrodes sintered by conventional sintering, infrared sintering and microwave sintering were investigated in detail. The polyol-synthesized AgNPs have a narrow particle size distribution with an average diameter of 43 nm and uniformly disperse in the solution. The dip-coating process by dip-coating 40 times and at $500\,\mu m\,s^{-1}$ of pulling rate are suitable to prepare silver film with good surface morphology (a thickness of 1.97 µm and a roughness of about 2 nm). Conventional sintering and infrared sintering can achieve low sheet resistance (less than $1\,\Omega\,sq^{-1}$) and good morphology of silver electrodes; however, high sintering temperature and long sintering time limit their application to some extent. The silver films obtained by microwave sintering have the lowest sheet resistance ($0.75\,\Omega\,sq^{-1}$) and a low surface roughness (less than 1 nm), and can achieve quick sintering of silver films (sintering time less than 1 min) to prepare silver electrodes. This preparation route could be applied to construct OLED, solar cells or other electronic devices.

Data accessibility. Data used to produce the results in figures 3, 8 and 12 are available within the Dryad Digital Repository: https://doi.org/10.5061/dryad.9s4mw6mbc [43].

Authors' contributions. T.C., S.B. and Y.Z. contributed to experiments and tests, and T.C. wrote this manuscript. H.Y. and X.G. contributed to the theoretical guidance and the decision of the experimental plan. All authors gave final approval for publication.

Competing interests. We declare we have no competing interests.

Funding. This study was funded by the National Key Research and Development Program (grant no. 2016YFB0401305).

Acknowledgements. The authors thank the editors and anonymous reviewers for their constructive suggestions.

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
