## [Reviewer comments · Royal Society Open Science]

Review History

RSOS-191571.R0 (Original submission)

Review form: Reviewer 1

Is the manuscript scientifically sound in its present form?

No

Are the interpretations and conclusions justified by the results?

Yes

Is the language acceptable?

No

Do you have any ethical concerns with this paper?

No

Have you any concerns about statistical analyses in this paper?

Yes

Recommendation?

Major revision is needed (please make suggestions in comments)

Comments to the Author(s)

This manuscript synthesized Ag NPs and silver film, and study the sintering on the conductivity of silver electrode. I suggest accepting this manuscript after a few questions are clarified.

1. What is atmosphere used for sintering?
2. What is the method used to measure the particle size? How to get the particle size of 62.2%? Please clarify in the manuscript.
3. What is the concentration of AgNO₃/EG solution?
4. What is the meaning of "PVP-coated" silver? How to confirm it?
5. 400 oC can completely remove the PVP? Please provide TGA data to confirm.
6. Please further clearly explain the effect of treating powder and treating time change on the resistance.
7. The author claimed that there no change of film surface morphology, what is reason of the sheet resistance decrease?
8. The language need to be polished.

Review form: Reviewer 2

Is the manuscript scientifically sound in its present form?

Yes

Are the interpretations and conclusions justified by the results?

Yes

Is the language acceptable?

Yes

Do you have any ethical concerns with this paper?

No

Have you any concerns about statistical analyses in this paper?

No

Recommendation?

Accept with minor revision (please list in comments)

Comments to the Author(s)

The paper involved the synthesis of Ag films, and it is very sound. So I think that it can be considered to be published after some English is revised.

Decision letter (RSOS-191571.R0)

19-Nov-2019

Dear Dr Guo:

Title: Facile preparation of high conductive silver electrodes by dip-coating followed by quick sintering

Manuscript ID: RSOS-191571

The editor assigned to your manuscript has now received comments from reviewers. We would like you to revise your paper in accordance with the referee and Subject Editor suggestions which can be found below (not including confidential reports to the Editor). Please note this decision does not guarantee eventual acceptance.

Please submit your revised paper before 12-Dec-2019. Please note that the revision deadline will expire at 00.00am on this date. If we do not hear from you within this time then it will be assumed that the paper has been withdrawn. In exceptional circumstances, extensions may be possible if agreed with the Editorial Office in advance. We do not allow multiple rounds of revision so we urge you to make every effort to fully address all of the comments at this stage. If deemed necessary by the Editors, your manuscript will be sent back to one or more of the original reviewers for assessment. If the original reviewers are not available we may invite new reviewers.

RSC Associate Editor:
Comments to the Author:
(There are no comments.)

RSC Subject Editor:
Comments to the Author:
(There are no comments.)

Reviewers' Comments to Author:

Reviewer: 1

Comments to the Author(s)

This manuscript synthesized Ag NPs and silver film, and study the sintering on the conductivity of silver electrode. I suggest accepting this manuscript after a few questions are clarified.

1. What is atmosphere used for sintering?
2. What is the method used to measure the particle size? How to get the particle size of 62.2%? Please clarify in the manuscript.
3. What is the concentration of AgNO₃/EG solution?
4. What is the meaning of "PVP-coated" silver? How to confirm it?
5. 400 oC can completely remove the PVP? Please provide TGA data to confirm.
6. Please further clearly explain the effect of treating powder and treating time change on the resistance.
7. The author claimed that there no change of film surface morphology, what is reason of the sheet resistance decrease?
8. The language need to be polished.

Reviewer: 2

Comments to the Author(s)

The paper involved the synthesis of Ag films, and it is very sound. So I think that it can be considered to be published after some English is revised.

Author's Response to Decision Letter for (RSOS-191571.R0)

See Appendix A.

RSOS-191571.R1 (Revision)

Review form: Reviewer 1

Is the manuscript scientifically sound in its present form?

Yes

Are the interpretations and conclusions justified by the results?

Yes

Is the language acceptable?

Yes

Do you have any ethical concerns with this paper?

No

Have you any concerns about statistical analyses in this paper?

No

Recommendation?

Accept as is

Comments to the Author(s)

The authors revised the manuscript accordingly. The current version can be published.

Decision letter (RSOS-191571.R1)

17-Dec-2019

Dear Dr Guo:

Title: Facile preparation of high conductive silver electrodes by dip-coating followed by quick sintering

Manuscript ID: RSOS-191571.R1

It is a pleasure to accept your manuscript in its current form for publication in Royal Society Open Science. The chemistry content of Royal Society Open Science is published in collaboration with the Royal Society of Chemistry.

RSC Associate Editor:
Comments to the Author:
(There are no comments.)

RSC Subject Editor:
Comments to the Author:
(There are no comments.)

Reviewer(s)' Comments to Author:

Reviewer: 1

Comments to the Author(s)

The authors revised the manuscript accordingly. The current version can be published.

Appendix A

Manuscript ID: RSOS-191571

Title: Facile preparation of high conductive silver electrodes by dip-coating followed by quick sintering

Author(s): Tianrui Chen, Hui Yang, Shengchi bai, Yan Zhang, Xingzhong Guo*

Response to Referee

Dear editor and referees,

Thank you very much for the constructive suggestions with regard to our manuscript. The reviewer's comments are all valuable and helpful for revising and improving our paper. We have revised the paper point by point according to their comments, and hope that the revised paper could meet the requirements of acceptance. Attached are the responses to the reviewers' comments, and the clean final version of my manuscript. The change part has been marked out in red font and yellow back.

If you have any questions about this paper, please don't hesitate to contact us.

I am looking forward to hearing from you.

Best regards,

Xingzhong Guo

Zhejiang University

Email: msewj01@zju.edu.cn

Reviewer Comments

Reviewer: 1

Comments:

This manuscript synthesized Ag NPs and silver film, and study the sintering on the conductivity of silver electrode. I suggest accepting this manuscript after a few questions are clarified.

Answer: Thank you very much for your constructive and valid comments.

Q#1. What is atmosphere used for sintering?

Answer: Three sintering processes were used to achieve the sintering of the dip-coated silver films after dip-coating. In the conventional sintering, the silver films were sintered in a muffle furnace at

different temperatures. In the infrared sintering, the silver films were placed under an infrared baking lamp and baked at different power for different sintering time. In the microwave sintering, the silver films were placed in a microwave processing instrument at different microwave powers for different sintering time. All three sintering processes are in air atmosphere.

Q#2. What is the method used to measure the particle size? How to get the particle size of 62.2%? Please clarify in the manuscript.

Answer: Thank you for your reminding. In this work, the diameters of AgNPs were measured and counted by *Nano Measurer* software. We measured 1000 particles and counted. It is demonstrated that the synthesized AgNPs has a small particle size and narrow particle size distribution, and the average size of silver nanoparticles is about 43 nm, and the particle size of about 62.2% is distributed between 31.5 and 54 nm.

Q#3. What is the concentration of AgNO₃/EG solution?

Answer: Thank you for your reminding. In the synthesis process, the concentration of AgNO₃/EG solution is 600 mM.

Q#4. What is the meaning of “PVP-coated” silver? How to confirm it?

Answer: Thank you for your reminding. After dip-coating, the dip-coated silver electrodes without any post-treatment were non-conductive, which indicates that the silver particles were coated by PVP. The composition of AgNPs was characterized by EDS and XRD, as shown in Figs. 1d and 1e. It could be seen that the content of silver reaches 83.77%, the contents of C and N elements are 13.69% and 2.55%, respectively, indicating that the existence of PVP in the samples.

Q#5. 400°C can completely remove the PVP? Please provide TGA data to confirm.

Answer: Thank you for your reminding. When the temperature further increases to 400 °C, the nanoparticles are completely deformed and wrecked, and there exist some isolated large silver particles or little silver bulk. It is seen from the TG and DTA of AgNPs (Fig. 3f) that a sharp exothermic peak near 300 °C is attributed to the decomposition of PVP, which suggests that the temperature of conventional sintering should be designed more than 300 °C to remove the PVP. It can be seen from the TG diagram that there is an obvious weight loss between 200 and 400°C, which should be due to the decomposition of PVP.

Q#6. Please further clearly explain the effect of treating powder and treating time change on the resistance.

Answer: Thank you for your suggestion. We add the explanation of the effect of powder and treating time on the resistance in the resubmitted manuscript.

Q#7. The author claimed that there no change of film surface morphology, what is reason of the sheet resistance decrease?

Answer: Thank you for your reminding. Fig. 8 shows the thickness and roughness of infrared sintered silver electrodes measured by step profiler. The thicknesses of silver films after infrared sintered at 50 W for 30 min, 200 W for 30 W and 200 W for 60 min are 1.94, 2.16 and 2.07 μm , and the roughness of the three samples is 1.3, 1.39 and 0.68 nm, respectively, indicating no significant change. It can also be considered that infrared sintering has no obvious effect on the surface morphology of the silver films, while can promote the deformation of PVP to increase the conductivity of silver films.

Q#8. The language need to be polished.

Answer: The language has been polished, and all the other corrections have been given in revised manuscript.